# Study on the Effect of Main Chain Molecular Structure on Adsorption, Dispersion, and Hydration of Polycarboxylate Superplasticizers

**DOI:** 10.3390/ma16134823

**Published:** 2023-07-04

**Authors:** Yunhui Fang, Zhanhua Chen, Dongming Yan, Yuliang Ke, Xiuxing Ma, Junying Lai, Yi Liu, Geli Li, Xiaofang Zhang, Zhijun Lin, Zhaopeng Wang

**Affiliations:** 1KZJ New Materials Group Co., Ltd., Xiamen 361101, China; fangyunhui@126.com (Y.F.); zhanhuachen123@126.com (Z.C.); yuliang_ke8207@163.com (Y.K.); mary@xmabr-kzj.com (X.M.); 15859120891@163.com (X.Z.); geniusvcap@163.com (Z.L.);; 2Polytechnic Institute, Zhejiang University, Hangzhou 310015, China; 3College of Civil Engineering and Architecture, Zhejiang University, Hangzhou 310058, China; junyinglai@zju.edu.cn; 4College of Materials Science and Engineering, Zhejiang University, Hangzhou 310023, China; liuyimse@zju.edu.cn

**Keywords:** main chain, polycarboxylate ether, adsorption, dispersion, hydration

## Abstract

Polycarboxylate ether (PCE) with different main chain structures was prepared by aqueous solution free radical polymerization using unsaturated acids containing sulfonic acid groups, acrylamide groups, and carboxyl groups and isoprenyl polyoxyethylene ether (IPEG). The molecular structure was characterized by infrared spectroscopy and gel chromatography, while adsorption, dispersion, and hydration properties were studied using a total organic carbon analyzer, rheometer, and isothermal microcalorimeter, respectively. The results show that the adsorption process of PCE on cement particles is spontaneous physical adsorption. The adsorption forces are mainly electrostatic interaction, and hydrogen bonding. The introduction of sulfonic acid groups and polycarboxylic acid groups reduces the initial adsorption amount of PCE but can accelerate the adsorption rate of PCE on cement and increase the adsorption amount at the adsorption equilibrium. The introduction of acrylamide groups in the PCE main chain is beneficial to the initial dispersion of PCE and can reduce the plastic viscosity of cement slurry. PCE can delay the hydration of cement. The introduction of acrylamide groups and dicarboxylic acid groups in the PCE main chain helps prolong the induction period of cement hydration, while the introduction of sulfonic acid groups is not conducive to its retarding effect.

## 1. Introduction

Concrete admixtures are a relatively effective and convenient technical approach to reducing cement consumption and improving the durability and strength of concrete. They are essential components of modern concrete and key new materials for the development of high-strength, high-performance, and high-durability concrete [1,2]. Polycarboxylate ethers (PCEs) have great potential for high performance due to their strong molecular structure adjustability and the ability to produce products with different properties using different monomer compositions and polymerization processes. They have become the most effective, economical, and convenient technical approach to achieving high durability and performance improvement in concrete, gradually replacing lignosulfonate-based superplasticizers, naphthalene-based superplasticizers, etc., and becoming the mainstream variety of concrete admixtures in the market [3].

The main chain molecular structure of PCE is generally composed of unsaturated acids and their derivatives, with sulfonic acid groups (-SO_3_H) and carboxylic acid groups (-COOH) grafted onto the main chain. The side chains are composed of long polyethylene oxide groups [4], which can ionize carboxylic acid groups, sulfonic acid groups, and other functional groups under alkaline conditions in the cement slurry, resulting in a negative charge on the main chain. PCE adsorbs on the surface of positively charged cement particles and hydration products through electrostatic forces and releases free water in the cement particles through the effects of steric hindrance and electrostatic repulsion, greatly increasing the fluidity of the slurry [5]. After PCEs are adsorbed onto cement particles, the interparticle forces are primarily electrostatic repulsion. However, in actual cement slurries, various high-concentration ions (such as Ca^2+^ and OH^−^) reduce the electrostatic repulsion, resulting in differences in the origin of repulsive forces between different cement particles.

The molecular structure of PCE can usually be modified. There are many studies on the molecular structure modification and working mechanism of PCE [6,7]. Different spatial hindrance effects can be produced by changing the length of the polyethylene oxide side chain or inserting various functional groups into the PCE skeleton.

Different functional groups and structures in PCE molecules can result in varying degrees of affinity to the surface for cement particles and cement hydration products, which in turn leads to significant adsorption. The adsorption process belongs to solid-liquid adsorption and is a dynamic equilibrium process in which adsorption and desorption occur simultaneously [8]. The large difference in size between solid and liquid phase molecules leads to significant differences in adsorption, which can be studied through adsorption kinetics and thermodynamics. Isothermal adsorption and adsorption kinetics are commonly used to simulate adsorption between solid and liquid phases, and commonly used isothermal and kinetic models are listed in Table 1 [9].

After different molecular structures of PCE are adsorbed onto the surface of cement particles or hydration products, they can have different effects on the rheological properties of fresh cement paste, such as monomer molar ratio, side chain length, and anionic anchoring groups. Yamada K et al. [4] analyzed and studied the effect of PCE side chain length, main chain polymerization degree, carboxyl and sulfonic acid group composition on dispersibility, and found that PCE with longer side chains, a lower main chain polymerization degree, and higher sulfonic acid group content had better dispersibility. HE Yan [10] found that the carboxyl density of PCE has a significant effect on its dispersibility, and the dispersibility of PCE increases with increasing carboxyl density, while the viscosity of cement paste is the lowest. Hui Feng [11] pointed out that PCE with short side chains and high carboxylate density has higher adsorption capacity in mortar, and PCE with sulfonic acid groups has higher adsorption on the surface of cement, resulting in better rheological properties.

PCE also has a huge impact on cement hydration through the free water released by the dispersed cement particles and the morphology and thickness of the PCE wrapping layer on the surface of the cement particles. In the different stages of hydration, the main sources of heat release are mineral dissolution, nucleation, and the growth of hydration products. Different microstructures of PCE and their interaction with cement particles in the early stage lead to different rates of mineral dissolution, pore solution concentration, nucleation, and growth rate of hydration products. Generally, the new surface film layer caused by adsorption may be thicker than the original one, and the presence of a thick film layer can hinder the exchange of substances and energy between the surface of cement particles and the liquid phase, and also hinder the penetration of external ions into the internal ions, thereby affecting the hydration process of cement. Hongwei Tian [12] studied its effect on the flowability and early hydration of Portland cement and sulphoaluminate cement systems and showed that PCE can significantly inhibit the early formation of ettringite (AFt) crystals. Karen L. Scrivener [13] studied the fundamental mechanisms of hydration during different stages and showed that the growth of the main hydration product, C-S-H, is the main factor controlling the main heat evolution peak, with geochemical dissolution theory as an explanation for the induction period.

The hydration heat release curve can be divided into five stages, including (I) Initial period, (II) Induction period, (III) Acceleration period, (IV) Deceleration period, and (V) Decline period. In the induction period, after water is added to cement, mineral phases, such as gypsum and C3A dissolve rapidly and generate AFt. At the same time, the concentration of Ca^2+^, OH^−^, SO_4_^2−^, and other ions in the cement solution continues to increase. When the ion concentration reaches a certain value, the nucleation and crystallization processes occur at the solid-liquid interface, resulting in the production of calcium hydroxide (CH) and AFt crystals. Meanwhile, tricalcium silicate (C3S) is gradually hydrated to form a small amount of short and fibrous hydrated calcium silicate gel (C-S-H). Therefore, it is difficult to observe a smooth induction period in ordinary Portland cement paste without the addition of PCE, and the hydration heat curve shows a sharp “valley” shape. The decrease in the rate and amount of heat released during the induction period can be explained by the double electric layer theory. This theory suggests that a calcium-deficient, silicon-rich layer is formed on the surface of cement particles, which can adsorb calcium ions from the solution and form a double electric layer. The appearance of this double electric layer can prevent further contact between cement particles and water, thereby delaying the hydration of cement and causing it to enter the induction period [14]. According to the crystal dissolution theory mentioned by Juilland, the duration of this initial reaction is short, and it is difficult to capture the heat release pattern of this stage [15,16].

Hydration calorimetry aims to study the exothermic behavior and final state of different hydration processes but lacks the main factors causing exothermic behavior at different stages. Hydration kinetics can characterize the states and reaction processes dominated by different reaction mechanisms during hydration. The Krstulovic–Dabic model for hydration kinetics is currently widely recognized and extensively used in the study of composite cementitious systems [17,18,19]. It considers three stages of the hydration process in cementitious materials: nucleation and crystal growth (NG), interfacial reaction (I), and diffusion (D). It is generally believed that during the NG and I stages, the hydration products continue to grow and are in an accelerated stage. Krstulovic and Dabic characterized these three hydration processes using the JMAK model [14], the Brown model, and the Jander model, respectively. The kinetic equations governing these three processes are as follow:(1)$$NG\text{:}\;{\left[-\ln\left(1-\alpha \right)\right]}^{\frac{1}{n}}=K_{NG}\left(t-t_{0}\right)$$
(2)$$I\text{:}\;{\left[{1-\left(1-\alpha \right)}^{1/3}\right]}^{1}=K_{I}\left(t-t_{0}\right)$$
(3)$$D\text{:}\;{\left[{1-\left(1-\alpha \right)}^{1/3}\right]}^{2}=K_{D}\left(t-t_{0}\right)$$

Hydration kinetics models have been applied in many aspects, such as studying the variation of n value and apparent activation energy during crystal growth in multiphase systems and investigating the dominant reactions in the hydration process of multiphase systems at different temperatures [16]. In blended cements with slag content exceeding 50%, interfacial reactions are the main reactions. Typically, hydration kinetics examine reaction parameters at different stages. The main parameters for the NG process are *K_NG_* and n, where *K_NG_* represents the rate constants for nucleation and crystal growth in the NG process, reflecting the speed of the hydration reaction. A larger *K_NG_* indicates easier reaction occurrence, leading to a significant increase in the nucleation rate of calcium silicate hydrate (C-S-H) and calcium hydroxide (CH) crystals. When the concentration of Ca^2+^ in the solution increases, C-S-H is more likely to reach a saturated state. Additionally, the NG process is influenced by the reaction order, where the magnitude of the reaction order represents the extent to which concentration affects the reaction rate. Therefore, the fitting curve of the NG process is a parabola.

Existing research on PCE has mainly focused on the effects of side chain structures, such as side chain length and side chain density, on performance, while there is limited research on the influence of main chain structures. In this study, a series of PCEs with different main chain molecular structures were synthesized and prepared. Mathematical models, including the isothermal adsorption model, adsorption kinetics, rheological model, and hydration kinetics were used to investigate the influence of main-chain molecular structure on the performance of PCEs. Furthermore, the correlation between different main chain molecular structures of PCEs and the performance of cement slurries was systematically elucidated, along with the mechanism of “adsorption-dispersion-hydration” interaction.

## 2. Materials and Methods

### 2.1. Materials

(1) The benchmark cement used in this study was produced by China United Cement Group Co., Ltd., (Beijing, China), and its chemical composition and physical properties are shown in Table 2. (2) Isopentenyl polyoxyethylene ether (IPEG, M_W_ = 2400 g·mol^−1^, average degree of polymerization was produced by Jiahua Chemical Co., Ltd. (Jinhua, China) acrylic acid (AA), methacrylic acid (MAA), sodium methacrylate sulfonate (SMS), 2-acrylamide-2-methylpropane sulfonic acid (AMPS), and itaconic acid (IA) were all of chemical purity and purchased from Sinopharm Chemical Reagent Co., Ltd. (Shanghai, China) ammonium persulfate (APS) was produced by Jinan Fengle Chemical Co., Ltd. (Jinan, China) Sodium hexametaphosphate (SH) was produced by Jining Shunyida Chemical Technology Co., Ltd. (Jining, China) deionized water (W) was used throughout the experiments. The chemical composition and physical properties of the cement are shown in Table 2.

### 2.2. Synthesis of PCE

TPEG, SH, and W were added into four-necked flasks and stirred with a magnetic stirrer until dissolved. Then, the unsaturated monomers, AA, and W were mixed and stirred until dissolved to obtain the A solution. APS and W were mixed to obtain the B solution. A and B solutions were uniformly drip-fed using a peristaltic pump at 60 °C for 3 h. After the drip-feeding, the mixture was kept at 60 °C for 1 h and then cooled down to 40 °C. The mixture was neutralized to pH 6–7 using a 32% NaOH solution. The synthesis route is shown in Figure 1, and the specific monomer ratios are presented in Table 3.

### 2.3. Test Methods

#### 2.3.1. FI-IR

The dried PCE was ground together with KBr and pressed into a thin film for testing using an Avatar 360 Fourier transform infrared spectrometer from Nicolet (Waltham, MA, USA).

#### 2.3.2. SCD

Specific charge density (SCD) of the PCE was measured using a particle charge detector (PCD05, Mütek Analytic, Frankfurt, Germany). The sample was diluted to a concentration of 0.1 wt% and then titrated with a polydiallyldimethylammonium chloride solution to completely neutralize the negative charge of PCE.

#### 2.3.3. GPC

The molecular weight, molecular weight distribution, intrinsic viscosity, and hydrodynamic radius of the sample were measured using a Waters 1515/2414 gel permeation chromatograph with a flow rate of 0.8 mL/min and a flow phase of 0.1 mol/L NaNO_3_ aqueous solution. The synthesized PCE was diluted to a concentration of 5 mg/mL.

#### 2.3.4. Adsorption Amount

An elementary TOC-VCPH total organic carbon analyzer was used to test the adsorption amount of PCE on cement. Different concentrations of PCE solutions were prepared, and 20 g of cement was added to 40 mL of PCE solution. After stirring evenly, an appropriate amount of liquid was taken and poured into a centrifuge tube. The upper, clear liquid was collected for TOC testing after centrifugal separation and filtration (5000 rpm, 10 min). The adsorption amount of PCE on cement was calculated according to Equation (4).
(4)$$\Gamma =\frac{\left(c_{0}\times V_{1}-c_{1}\times V_{1}\right)}{m\times 1000}$$

#### 2.3.5. Fluidity

In this study, a fluidity test of fcps (fresh cement pastes) was carried out to investigate the dispersing performance of PCE according to GB/T 8077-2012 “Test Method for Homogeneity of Concrete Admixtures” [20]. The water-to-cement ratio was 0.29.

#### 2.3.6. Rheology

PCE was added to the fcps at a dosage of 0.25 mg/g cement, with a water-to-cement ratio of 0.29. The rheological properties of the fcps were measured using a rotational rheometer (RheolabQC, Anton Paar, North Ryde, NSW, Australia). First, the sample was pre-sheared for 30 s at a speed of 5 r/min to minimize the effect of thixotropy. The shear rate was increased from 0 to 131 rpm for 1 min, then held at 131 rpm for 18 s, and decreased from 131 to 0 rpm for 1 min. The environmental temperature and humidity were maintained at 20 ± 2 °C and 60 ± 5%, respectively. Three tests were performed for each cement slurry, the specific testing procedure is shown in Figure 2.

The yield stress and viscosity were calculated using the Bingham model (5) and Herschel-Bulkley model (6). For shear-thinning mixtures, *n* < 1, and for shear-thickening mixtures, *n* > 1.
(5)$$\tau =\tau_{0}+\mu_{0}\gamma$$
(6)$$\tau =\tau_{0}+\mu_{0}\gamma^{n}$$
where *τ* is the shear stress (Pa); *τ*_0_ is the yield stress (Pa); *μ*_0_ is the plastic viscosity (mPa·s); *γ* is the shear rate (s^−1^); and *n* is the rheological behavior index.

#### 2.3.7. Hydration

In this study, the hydration heat curve of cement was measured using a TAM Air isothermal calorimeter (TA, Waters Corporation, Newark, DE, USA). Before the experiment, the sample and deionized water were placed in a constant-temperature chamber at 25 °C for 24 h. The water-to-cement ratio was 0.35, and 100 g of cement and 35 g of PCE aqueous solution were weighed and thoroughly mixed. Then, 3.00 g of cement slurry was weighed and tested using the TAM Air isothermal calorimeter.

## 3. Results and Discussion

### 3.1. Molecular Structure Characterization

#### 3.1.1. FI-TR

The characterization of the PCE molecule structure using Fourier transform infrared spectroscopy (FT-IR), and the test results are shown in Figure 3.

From Figure 3, it can be seen that all PCE samples exhibit characteristic peaks at 2867 cm^−1^, 1729 cm^−1^, 1454 cm^−1^, 1107 cm^−1^, and 946 cm^−1^. The absorption peak at 2867 cm^−1^ is due to the stretching vibration of C-H bonds in the alkane group of the side chain of the polyethylene oxide (PEO) chain. The absorption peak at 1729 cm^−1^ is due to the symmetrical stretching vibration of the C=O bond in the carboxylic group (-COOH). The strongest absorption peak near 1107 cm^−1^ is due to the asymmetric stretching vibration of the C-O-C bond in the ether group. The absorption peak at 946 cm^−1^ is due to the out-of-plane bending vibration of the trans double bond of the OH group. These results indicate that the PCE molecules with different main chain structures synthesized in this study all contain PEO chains, carboxylic groups, ether bonds, and hydroxyl groups. In addition, a characteristic peak of the sulfonic acid group was observed at 1350 cm^−1^ in the FT-IR spectra of PCE-3 and PCE-4, indicating that the molecular structures contain sulfonic acid groups as expected in the design, which confirms the successful synthesis of PCE.

#### 3.1.2. SCD

After adsorption onto the surface of cement and other mineral phases by adsorption, PCE exhibits various molecular conformations. The further dispersion of the cement mineral phase system is achieved through the steric hindrance effect of the side chain. Therefore, the study of the solution conformational behavior and intrinsic charge density of PCE is of great significance for the in-depth exploration of the mechanism of PCE’s action on cement. The particle charge meter was used to test the charge density of different concentrations of PCE, and the results are shown in Figure 4.

Figure 4 shows the charge density of PCE (concentration of 800 mg·L^−1^). It can be seen that at a PCE concentration of 800 mg·L^−1^, the order of charge density is PCE-5 > PCE-4 > PCE-3 > PCE-2 > PCE-1. This indicates that the charge density of PCE has been improved to some extent by using different functional monomers to replace AA. Among them, PCE-5 has the highest charge density mainly because it introduces itaconic acid into the main chain, which contains two carboxyl functional groups, resulting in a high degree of ionization and a high charge density.

#### 3.1.3. GPC

The differences in the molecular structure of PCE (such as side chain length, side chain density, charge density, molecular weight, etc.) will have different effects on the workability of cement slurry [21]. Gel permeation chromatography (GPC) was used to characterize the molecular structure of the synthesized PCE, where the calculation formulas for carboxyl density (D_AA_) and main chain polymerization degree (L_bb_) are shown in Equations (7) and (8).
(7)$$D_{\mathrm{AA}}=\frac{n_{AA}+n_{\left(fuction\;monomer\right)}}{\begin{array}{c} n_{AA}+n_{TPEG}\cdot Z_{z} \end{array}}$$
(8)$$L_{\mathrm{bb}}=\frac{M_{w}}{\begin{array}{c} M \end{array}}$$

D_AA_ represents side chain density; L_bb_ represents the main chain polymerization degree; *Z_Z_* represents the conversion rate of TPEG monomer; *n_TPEG_* represents the amount of TPEG material during feeding; *n_AA_* represents the amount of AA material during feeding, *n _(functional monomer)_* represents the amount of substance replacing AA, where itaconic acid is a dioic acid and the amount of substance replacing AA needs to be multiplied by a coefficient; *M_w_* represents the weight-average molecular weight of PCE; *M* represents the theoretical molecular weight of structural units. The test results are shown in Table 4.

From Table 4, it can be observed that the number-average molecular weight (M_n_) and weight-average molecular weight (M_w_) of PCE vary greatly, and the molecular weight distribution of PCE is relatively small, with all values below two. PCE-3 has the smallest molecular weight due to the chain transfer effect of the SMS used in the PCE-3 polymerization reaction, which reduces the molecular weight of PCE.

The Mark–Houwink α value of PCE-1 is 0.26. When sulfonic acid groups and carboxyl groups are introduced into the main chain of PCE, the Mark–Houwink α value decreases, indicating an increase in the curling degree of PCE molecules. PCE-3, which contains sulfonic acid groups, has the lowest Mark–Houwink α value of 0.16, indicating the highest degree of curling in solution. The changing trend is opposite to the change in charge density, indicating that the higher the density of negatively charged adsorption groups in PCE molecules, the more likely the molecules are to curl. A molecular configuration diagram of PCE was constructed based on infrared spectroscopy, GPC, and charge density data, as shown in Figure 5.

### 3.2. Adsorption Behavior

#### 3.2.1. Adsorption Isotherms

Carboxylic groups on the main chain of PCE carry negative charges and can adsorb onto the positively charged surfaces of cement particles and cement hydration products through Coulombic interactions. Meanwhile, carboxylic groups can also adsorb onto the surfaces of cement particles and cement hydration products by chelating with Ca^2+^. -SO^3−^ can also adsorb onto the positively charged surfaces of cement particles and cement hydration products through interactions. To investigate the effect of different main chain molecular structures of PCE on adsorption performance, the initial adsorption amounts of PCE were tested at concentrations of 0.5 mg·L^−1^, 1.0 mg·L^−1^, 1.5 mg·L^−1^, 2.0 mg·L^−1^, and 2.5 mg·L^−1^, respectively. The results are shown in Figure 6.

As shown in Figure 6, the adsorption capacity of PCE increases with increasing concentration. At lower concentrations (<0.5 mg·L^−1^), the adsorption capacity of all five types of PCE exhibits a sharp increase. During this period, the surface of cement particles becomes more active and contains more pores, resulting in higher adsorption rates. At higher concentrations (>1 mg·L^−1^), the adsorption capacity increases at a slower rate and gradually reaches a steady plateau.

To better determine the characteristic adsorption of PCE, we used Langmuir and Freundlich models to fit and analyze the data in Figure 6. The results are shown in Table 5.

From Table 5, it can be observed that the adsorption models of PCE with different main chain molecular structures have R^2^ values ranging from 0.9444 to 0.9997, indicating a high correlation. The Freundlich adsorption model shows even higher R^2^ values with n_F_ > 1, suggesting that PCE primarily undergoes monolayer adsorption. This is consistent with the monolayer adsorption characteristic of the Langmuir model. The order of saturation adsorption capacity (q_m_) is as follows: PCE-2 > PCE-1 > PCE-4 > PCE-3 > PCE-5. It can be observed that the adsorption amounts of PCE-3 and PCE-4 on cement are smaller than those of PCE-1, indicating that the introduction of sulfonic acid groups decreases the adsorption capacity of PCE. This could be attributed to the inability of PCE with sulfonic acid groups to adsorb onto the negatively charged cement particle surfaces through Ca^2+^ complexation, resulting in a decrease in saturation adsorption capacity. Furthermore, from Table 5, it can be noted that PCE-2 has a higher adsorption amount than PCE-1, indicating a significant increase in adsorption capacity upon the introduction of methacrylic acid. This could be attributed to the enhanced electronegativity of PCE molecules due to the introduction of methacrylic acid (as evidenced by the test results in Section 3.1.2). The increased content of negatively charged COO^−^ groups in the main chain of PCE enhances its ability to adsorb on the positively charged hydration product surfaces through electrostatic attraction, leading to an increase in the adsorption amount on the reference cement surface. On the other hand, PCE-5 exhibits a lower adsorption amount than PCE-1, suggesting a decrease in adsorption capacity upon the addition of itaconic acid. This could be attributed to the excessive electronegativity of PCE due to the introduction of the dicarboxylic acid groups on the main chain, causing the carboxyl groups on the molecule to be shielded by the random coil conformation of the polymer chain. As a result, the electrostatic repulsion and steric hindrance forces are weakened, leading to a decrease in adsorption on the cement surface.

The negative values of the adsorption-free energy ∆G for PCE indicate favorable and spontaneous adsorption onto cement particles. Generally, the absolute value of the change in ∆G for physical adsorption is smaller than that for chemical adsorption, with a range of −40 kJ·mol^−1^ to 0 kJ·mol^−1^ and −400 kJ·mol^−1^ to −80 kJ·mol^−1^, respectively. The ∆G values for PCE adsorption onto cement particles range from −27.5204 kJ·mol^−1^ to −25.4352 kJ·mol^−1^, indicating that physical adsorption is the dominant mechanism.

#### 3.2.2. Adsorption Kinetics of PCE

The adsorption rate of PCE on the surface of cement particles under isothermal conditions is mainly controlled by the following three processes [22]: (1) the movement rate of adsorbate molecules in the adsorption particle surface liquid film; (2) the diffusion rate inside the particles; and (3) the adsorption rate on the internal pore surface of the particles. To further investigate the effect of PCE with different main chain molecular structures on the adsorption rate, the adsorption amount of PCE-1~PCE-5 on the surface of cement particles was tested over time, and the results are shown in Figure 7.

As can be seen from Figure 7, the change in the adsorption amount of PCE with different main chain molecular structures on the surface of cement particles over time is similar. In the beginning, the adsorption is rapid, and with the extension of time, the adsorption amount gradually saturates and finally tends to equilibrium. Specifically, rapid adsorption occurs within the first 20 min, and the growth rate of the adsorption amount slows down after 20 min. After 100 min, the adsorption amount remains almost unchanged, indicating that the adsorption process has reached a kinetic equilibrium where the adsorption and desorption rates are equal.

The adsorption kinetics can not only reflect the adsorption rate of PCE on the cement surface, but also reveal its adsorption mechanism. The quasi-first-order (PFO) rate equation and the quasi-second-order (PSO) rate equation were used to fit the adsorption kinetics of PCE on the surface of cement particles, and the characteristic parameters of the fitting are shown in Table 6.

The high correlation coefficient R^2^ obtained from the PSO kinetic model in Table 6 indicates that the PSO adsorption kinetic equation can better describe the entire adsorption process of PCE on cement, which includes solution diffusion, liquid film diffusion, particle internal diffusion, and adsorption processes. Additionally, as shown in Figure 6, the PCE adsorption curve can be approximated by three segments, with a steeper slope for the 0–20 min segment, possibly due to solution and liquid film diffusion, a decreasing slope for the 20–100 min segment, possibly due to particle internal diffusion, and a plateau after 100 min, possibly due to pore adsorption.

It is also noteworthy that for PCE-1 and PCE-5, the adsorption amount slowly increases with time beyond 120 min, possibly due to the hydration effect, where some of the free PCE in the solution is adsorbed on the surface of hydration products. The fitting results of the internal diffusion model in Table I-D show poor correlation (<0.9) and curves that do not pass through the origin, indicating that internal diffusion is not the only rate-controlling step in PCE adsorption and other adsorption mechanisms may be involved [23]. Therefore, it is inferred that PCE adsorption on cement involves both physical adsorption (as demonstrated in Section 3.2.1) and chemical adsorption (hydration process), but physical adsorption is believed to be the main control mechanism for PCE adsorption on cement, as the hydration process is much longer than the diffusion process.

The equilibrium adsorption amount (q_e_) data indicate that PCE-2, PCE-3, and PCE-4 have higher qe values than PCE-1 and PCE-5, suggesting that introducing MAA, SMS, or AMPS into the main chain of PCE reduces its initial adsorption amount on cement but increases its adsorption amount at dynamic equilibrium. This is mainly because the amount of positively charged hydration products produced by cement during the hydration reaction increases as the adsorption time becomes longer. Based on the isothermal adsorption and adsorption kinetic data, the adsorption mechanism of PCE on the cement surface is shown in Figure 8.

### 3.3. Dispersion Behavior

#### 3.3.1. Fluidity

The water-reducing effect of PCE in freshly mixed concrete depends on its dispersibility and dispersion retention in the cement paste, which is closely related to the molecular structure of PCE. The flowability of cement paste containing PCE with different main chain molecular structures was tested, and the results are shown in Figure 9.

As shown in Figure 8, replacing acrylic acid-containing carboxyl groups with unsaturated substances containing sulfonic acid groups, amide groups, and other functional groups resulted in significant changes in the dispersibility of the cement paste. The order of the flowability of the cement paste is PCE-4 > PCE-2 > PCE-1 > PCE-5 > PCE-3. The initial flowability of PCE-3 is the smallest, which may be due to the smaller hydrodynamic radius (R_h_) of its molecules, resulting in less steric hindrance when adsorbed onto cement particles, thus reducing its dispersibility [24]. The adsorption amount of PCE-5 is less than that of PCE-1, indicating that introducing carboxyl groups into the main chain is detrimental to the dispersibility of the water reducer. This is mainly because the carboxyl density is too high, and the carboxyl groups of different water-reducing agents adsorbed on cement particles bridge after complexing with calcium ions in the solution, resulting in decreased dispersibility [25]. The dispersibility of PCE-2 and PCE-4 is improved compared to that of PCE-1, with PCE-4 showing the strongest dispersibility. This may be because the acrylamide group in the molecule can form hydrogen bonds with water molecules and polyethylene oxide groups in the liquid phase, thereby enhancing the steric hindrance. The H^+^ in the acrylamide group can easily dissociate into protons and exhibit acidity, which can interact well with the alkaline cement system, facilitating wetting and relative lubrication of the cement particle surface. The dispersing mechanism of PCE is shown in Figure 10.

#### 3.3.2. Rheological Behavior

The structural characteristics of cement paste can be characterized by the rheological properties of the paste. Rheological parameters are important indicators for characterizing the workability and flowability of fresh cement paste. Changes in the internal structure can be analyzed by rheological parameters, such as shear rate, shear stress, and viscosity. The shear stress and plastic viscosity of cement paste containing PCEs with different main chain molecular structures under different shear rates are shown in Figure 11a,b, respectively, when the water-to-cement ratio is 0.29. PCE dosage is 0.20% of the cement amount (based on solid content), and the data were fitted using the Bingham and Herschel–Bulkley models. The results are shown in Table 7.

From Figure 11 and Figure 12, it can be observed that the shear stress of different main-chain molecular structures of PCE increases with the increase in shear rate, showing a nearly linear relationship. At the same shear rate, the order of shear stress values from large to small is: PCE-3 > PCE-5 > PCE-1 > PCE-4 > PCE-2. At low shear rates (0–20 s^−1^), the plastic viscosity of PCE-3 drops sharply, exhibiting plastic fluid behavior, while the plastic viscosity of PCE-1 and PCE-45 changes slightly with the increase in shear rate. As the shear rate (20–130 s^−1^) increases, the plastic viscosity continues to decrease and tends to stabilize.

Equations fitted by the Bingham model and the H-B model for the shear rate-shear stress curve of the slurry both have high correlations, with correlation coefficients greater than 0.99. However, τ_0_ fitted by the H-B model is mostly negative, indicating that the rheology of the cement slurry with different main-chain molecular structures belongs to the Bingham fluid model.

The thixotropy of the slurry can be characterized by the size of the thixotropic hysteresis loop area. The upward and downward segments of the shear stress-shear rate curve of the cement slurry with different main-chain molecular structures added to form a closed loop, and the hysteresis loop areas are relatively large, indicating that the thixotropy of the cement slurry added with PCE is good [26]. The ranking of the size of thixotropic hysteresis loop areas is PCE-1 > PCE-3 > PCE-5 > PCE-2 > PCE-4. The cement slurry added with PCE-1 has the best thixotropy, indicating that the PCE particles form weak flocculation. When the flocculation structure is disrupted, the viscosity decreases, and the thixotropy increases. The introduction of other functional groups into the main chain weakens the thixotropy and increases the viscosity, resulting in better water retention properties in concrete.

### 3.4. Hydration Behavior

#### 3.4.1. Hydration Processes

The cement hydration process can be divided into five stages: (I) initial period, (II) induction period, (III) acceleration period, (IV) deceleration period, and (V) decline period (as shown in Figure 12a) [27]. The hydration heat of cement with different PCE additives was tested, and the hydration heat flow and cumulative heat curves are shown in Figure 12b. Characteristic parameters of the hydration heat curve were extracted, as shown in Table 8. The parameters t_0_, t_1_, t_2_, t_3_, and t_4_ represent the end time of the induction period, the start time of Stage I, the time of maximum slope in the acceleration period, the end time of the acceleration period, and the start time of Stage II, respectively. The parameters q_0_, q_2_, q_3_, Q_0_, Q_3_, and Q_0-3_ represent the heat release rate and cumulative heat at corresponding points. The parameter K_2_ is the transversal slope between 0 and 2, indicating the nucleation rate during the acceleration period.

As shown in Figure 12, the addition of PCE to cement samples delays the hydration of cement to varying degrees. During the induction period, the rate of dissolution of cement minerals decreases, extending the supersaturation required for the crystallization and precipitation of hydration products. The time required for hydration products to reach the supersaturation necessary for nucleation and growth is determined by the length of the induction period [28], which typically lasts for 2–4 h, with the end of the induction period being the initial setting time. Compared with the blank sample, the addition of PCE results in a longer time for the end of the induction period (t_0_), indicating that PCE prolongs the hydration time of cement. This is because the adsorbed PCE molecules hinder ion diffusion, thus inhibiting hydration during the induction period [29]. For PCE-5 with the highest carboxyl group density, its t_0_ is slightly larger than that of PCE-1, while the hydration rate q0 is the same as PCE-1, and Q_0_ is slightly larger than PCE-1, indicating that carboxyl group density has little effect on the degree of hydration during the induction period. For PCE-2 and PCE-3, t_0_ is smaller than the comparison sample PCE-1, which does not add functional monomers, and PCE-3 has the smallest t_0_ because the introduced SMS has the smallest molecular weight, the fastest adsorption rate, and is easily covered by hydration products, resulting in a lack of retarding effect. Although the dispersibility of PCE-4 is not as good as that of PCE-1, its molecular weight is the largest, resulting in a higher coverage of the cement particle surface and thus a higher retarding effect during the induction period.

From Figure 12b, it can be seen that during the acceleration period, the rate of heat release from hydration increases with increasing hydration time. However, this rate does not increase linearly, but rather exhibits a maximum heat release rate acceleration, which is related to the nucleation and growth rate of cement. According to Table 8, after the addition of the water reducer, both t_2_ and K_2_ are greater than the blank, indicating that during the acceleration period, PCE affects the change in ion concentration and influences the nucleation and growth of hydration products. t_3_ > blank, while q_3_ < blank. The reason for this is that at the beginning of the acceleration period, the blank sample has less free water and thus a lower capacity for supersaturated solution, resulting in faster consumption of Ca^2+^. On the other hand, there is no hindrance from PCE on the surface of cement particles and hydration products, leading to rapid nucleation and growth. The k_2_ and q_3_ of PCE-5 are larger than those of PCE-1, due to its higher carboxyl density and better adsorption performance, as the bidentate carboxyl groups play an anchoring effect. The PCE in the solution is consumed faster, leading to less hindrance to nucleation and the growth of hydration products. The q_3_ values of PCE-2, PCE-3, and PCE-4 are all larger than those of PCE-1, indicating that at the end of the acceleration period, the nucleation and growth rates of hydration products reach their maximum. PCE-4 also benefits from its sulfonic acid group, which has a higher adsorption capacity, and the acrylamide group, which hydrolyzes under alkaline conditions, increasing its electronegativity and promoting cement hydration during the acceleration period. PCE-2, due to the presence of methyl groups, also promotes hydration during the acceleration period. PCE-3 has a lower molecular weight and a weaker water-reducing effect than PCE-1, so it shows a promotion effect on hydration during the acceleration period compared to PCE-1.

#### 3.4.2. Hydration Kinetics

Based on the Krstulovic–Dabic model, a hydration reaction kinetics model of the benchmark cement system was established using isothermal microcalorimetry. The parameter changes of the kinetic model were investigated for different main chain molecular structures, and a functional relationship between the degree of hydration and the hydration rate was established. By fitting the hydration process of the benchmark cement system, the impact of PCE on hydration products and cement hydration mechanisms, as well as the controlling factors of different hydration stages, were revealed. The alpha-dalpha/dt curves of blank samples and PCE with different main chain functional groups during the hydration reaction were shown in Figure 13, and the extrapolation equations and hydration kinetic parameters of each sample were summarized in Table 9.

From Figure 13, it can be seen that the three hydration processes are fitted to the parameters at different stages of cement hydration. During the hydration reaction, all three processes exist simultaneously, but when the hydration reaches a certain stage, the proportion of each process involved in the reaction is different. Therefore, there is a dominant process at different stages of hydration, which is reflected in the slowest hydration rate of the dominant process. At the beginning of the acceleration period, the NG process is dominant because the particle size of its hydration products is too small, and the I process and D process can be ignored.

The main parameters of the N_G_ process are K_NG_ and n, where K_NG_ is the constant of crystallization and crystal growth rate of the N_G_ process, reflecting the speed of the hydration reaction. The larger K_NG_ is, the easier the reaction occurs, resulting in a significant increase in the rate of formation of C-S-H and Ca(OH)_2_ crystals. Therefore, the NG process can examine the nucleation and growth of hydration products during the acceleration period. For PCE-5, which has a higher carboxyl density than PCE-1, the K_NG_ of PCE-5 is similar, but the reaction index n is significantly different. The n value of PCE-1 is 2.01, indicating that most of the hydration products are flocculent C-S-H or plate-like products during the acceleration period. However, PCE-5 has needle-like C-S-H and CH products. For PCE-2 or PCE-3, which contain amide or sulfonic acid groups, K_NG_ is smaller than that of PCE-1 without functional monomers, and the reaction index n is also smaller than that of PCE-1, indicating that the addition of functional groups can promote the increase of needle-like products. For PCE-4 containing both acrylamide and sulfonic acid groups, K_NG_ is the smallest, indicating that the hydration rate of acrylamide functional groups plays a promoting role.

As the molecular weight decreases, K_2_ changes little, indicating that the PCE molecular weight has little effect on the nucleation rate of hydration products at the beginning of the acceleration period [30]. The trend of the nucleation rate K_2_ analysis results is similar to that of the NG process nucleation and crystal growth rate constants. It is inferred that the K_NG_ of the NG process can better reflect the nucleation and crystal growth rates of C-S-H during the acceleration period. For the two samples with different carboxyl densities, PCE-5 has a larger q_3_ than PCE-1, consistent with the rule of a larger charge density and nucleation amount. PCE-1 has a smaller nucleation amount and requires a longer time for the hydration products to completely contact the phase interface, thus delaying the acceleration period of hydration [14]. For samples with different functional groups added, their maximum heat release rates are all greater than those of PCE-1. It is inferred that the maximum heat release rate at the end of the acceleration period is consistent with the nucleation amount estimated by the hydration kinetics of the NG processes K_NG_ and n. The NG process can accurately determine the change in nucleation sites before and after crystal growth, thus affecting the change in nucleation amount.

As time progresses, the hydration product layer becomes thicker, making the reaction more difficult, and the hydration control factor begins to shift towards the diffusion reaction rate D [15]. From the figure, it can be seen that the D process fits well with the hydration deceleration period and the stable period, and slightly underestimates the actual hydration rate for the deceleration period, but can still reflect the hydration reaction process well. The K_I_ is about 5 times that of K_D_, representing a much slower diffusion rate between particles than the NG and IPB processes, with very little liquid phase involved, and the influence of K_NG_ can be ignored. Upon entering the D process, the hydration degree required for the PCE-1 sample is equivalent to that of PCE-5, but greater than that of other samples. It is speculated that different functional groups still have an effect on the hydration rate in the later stage of the acceleration period, thereby affecting the time to enter the D process. The K_D_ values are basically the same, indicating that PCE has little effect on the D process.

## 4. Conclusions

Based on the study of the microstructure, adsorption, dispersion, and hydration of the main chain groups of PCE, the following conclusions were drawn:PCE exhibits a curled spherical structure in a neutral solution. The introduction of sulfonic acid and acrylamide groups on the PCE molecular main chain increases the anionic charge in the water solution and the degree of curling. PCE with sulfonic acid groups has the most curled configuration in solution.The introduction of sulfonic acid groups into the main chain of PCE reduces the saturation adsorption capacity, while the substitution of methacrylic acid with an equimolar amount of acrylic acid on the main chain increases the saturation adsorption capacity. Conversely, the substitution of acrylic acid with an equimolar amount of itaconic acid on the main chain decreases the saturation adsorption capacity.The adsorption process of PCE on cement particles is a spontaneous physical adsorption. The adsorption free energy ranges from −27.52 kJ·mol^−1^ to 25.43 kJ·mol^−1^. The main forces involved in the adsorption are electrostatic forces and hydrogen bonding. The introduction of sulfonic acid groups and multiple carboxylic acid groups leads to a decrease in the initial adsorption amount of PCE.The introduction of sulfonic acid groups increases the adsorption capacity of PCE at equilibrium and the rate of adsorption on cement particles, resulting in a faster attainment of adsorption equilibrium.The introduction of acrylamide groups on the PCE molecular main chain facilitates the initial dispersion of PCE and reduces the plastic viscosity of cement slurry.PCE can delay the hydration of cement. The introduction of acrylamide groups and binary carboxylic acid on the PCE molecular main chain is beneficial for extending the induction period of cement, while the introduction of sulfonic acid groups is not conducive to its retardation effect. This is related to the good electrostatic adsorption and spatial dispersion abilities of cement particles.PCE-1 has the smallest nucleation amount and the longest acceleration period, with the smallest maximum heat release peak, which is most effective in delaying hydration. PCE-5 has a high carboxyl density, and the hydration products during the acceleration period are mainly needle-shaped. However, PCE-4 has a negative effect on the NG process due to the presence of acrylamide groups.

## Figures and Tables

**Figure 1 materials-16-04823-f001:**
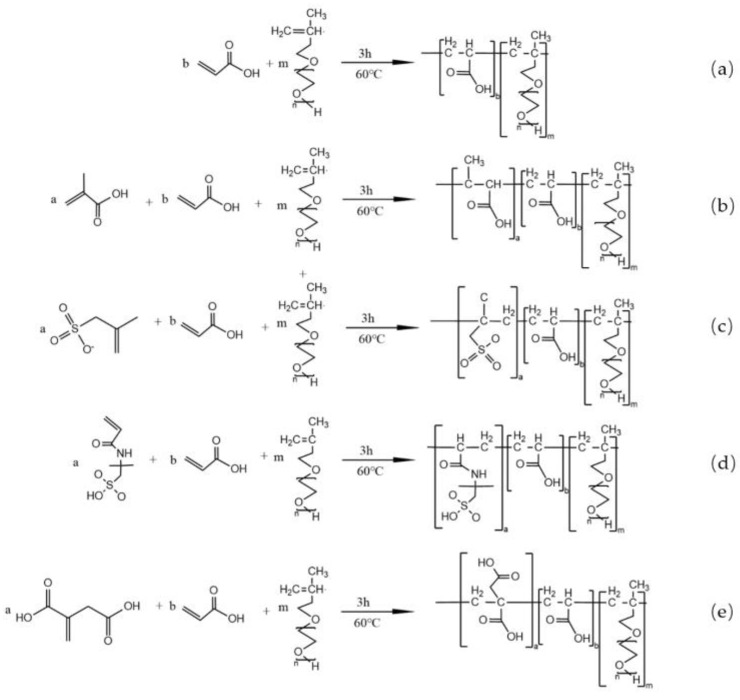
Synthesis route of PCE: (**a**) PCE-1; (**b**) PCE-2; (**c**) PCE-3; (**d**) PCE-4; (**e**) PCE-5.

**Figure 2 materials-16-04823-f002:**
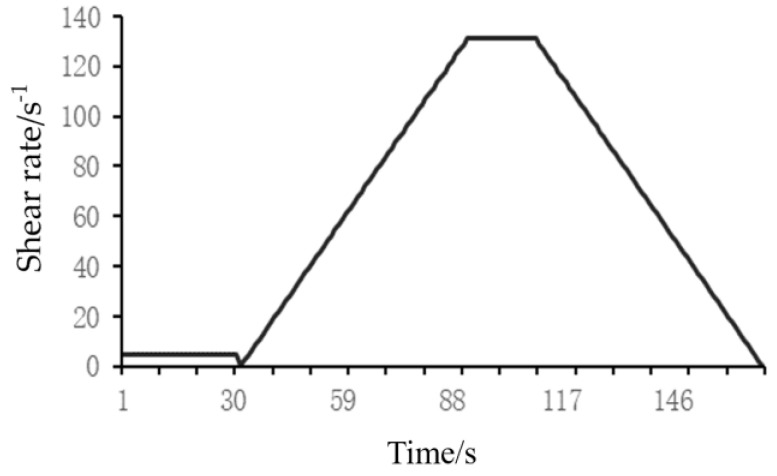
Rheological testing procedure for fcps.

**Figure 3 materials-16-04823-f003:**
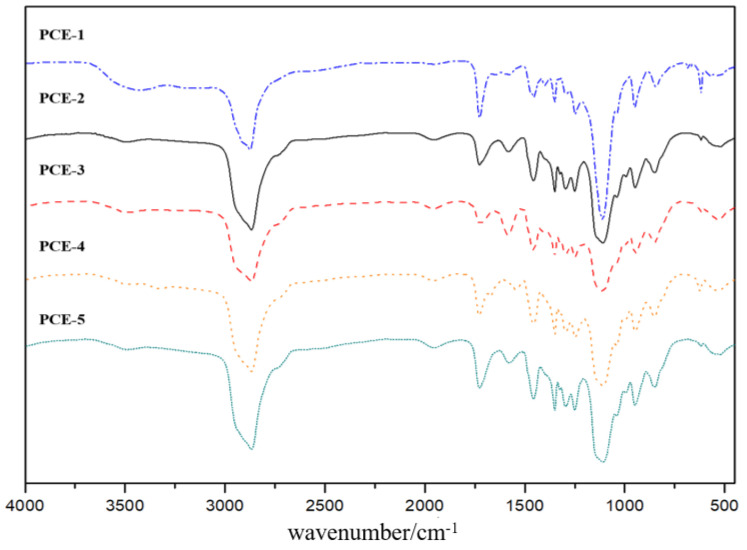
FT-IR spectra of PCEs.

**Figure 4 materials-16-04823-f004:**
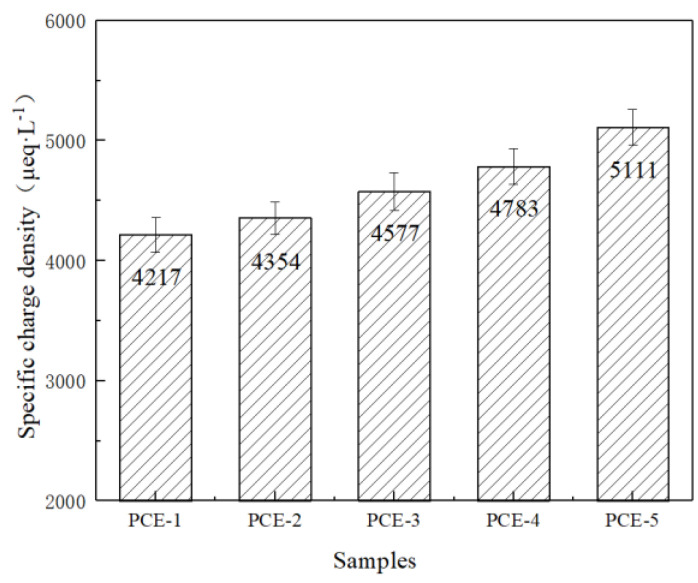
Specific charge density of PCE.

**Figure 5 materials-16-04823-f005:**
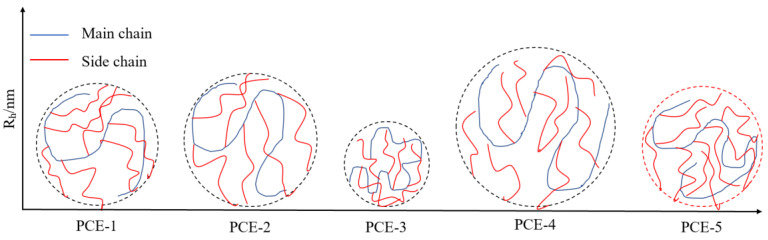
Molecular configuration of PCE.

**Figure 6 materials-16-04823-f006:**
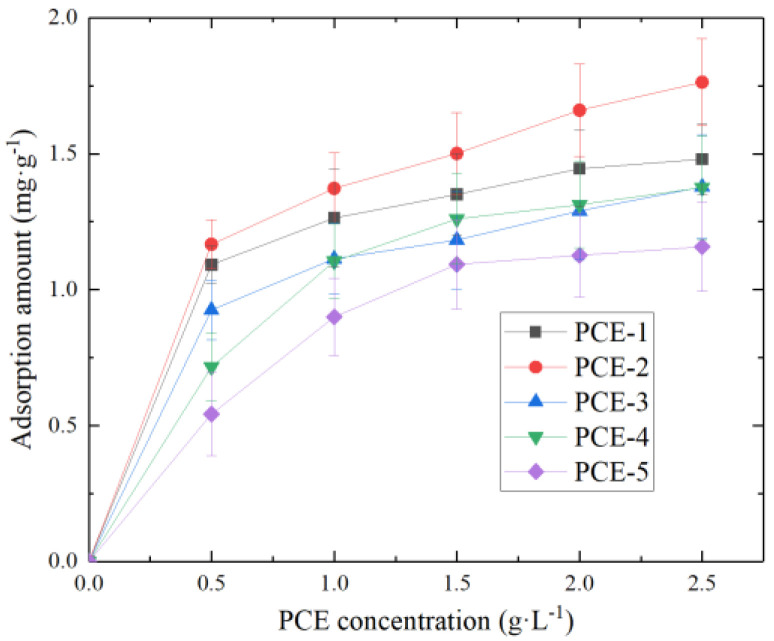
The adsorption amounts of PCE at different concentrations.

**Figure 7 materials-16-04823-f007:**
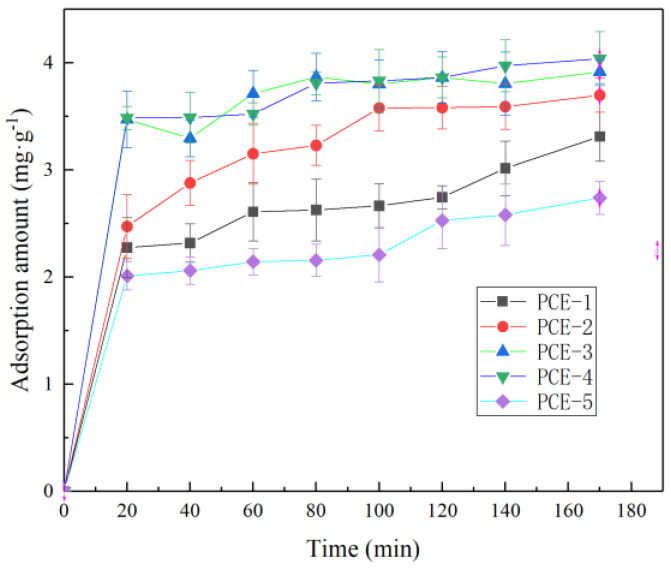
Adsorption kinetics curve of PCE on cement particles.

**Figure 8 materials-16-04823-f008:**
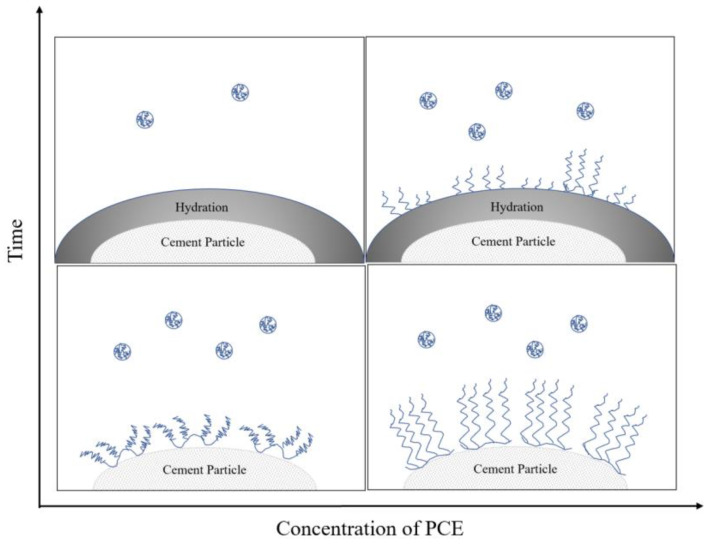
Adsorption mechanism of PCE on cement.

**Figure 9 materials-16-04823-f009:**
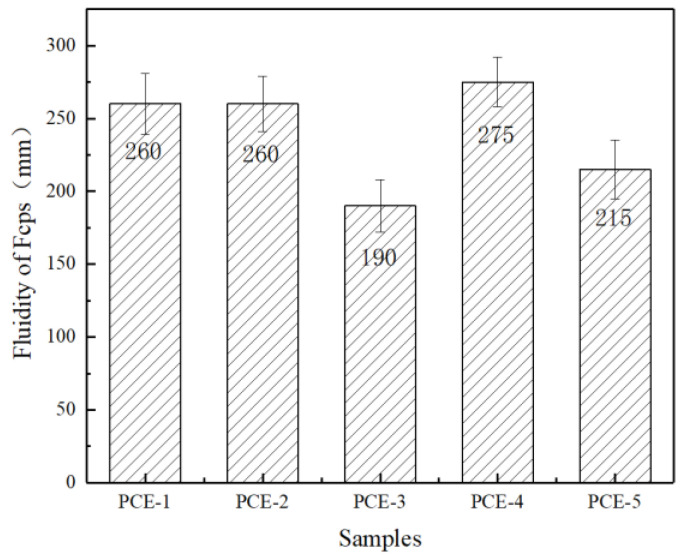
Influence of PCE on the fluidity of cement pastes.

**Figure 10 materials-16-04823-f010:**
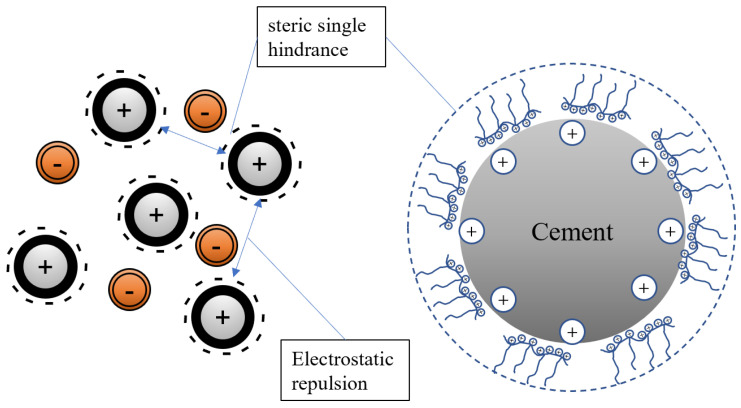
Scheme of interaction between PCE and cement.

**Figure 11 materials-16-04823-f011:**
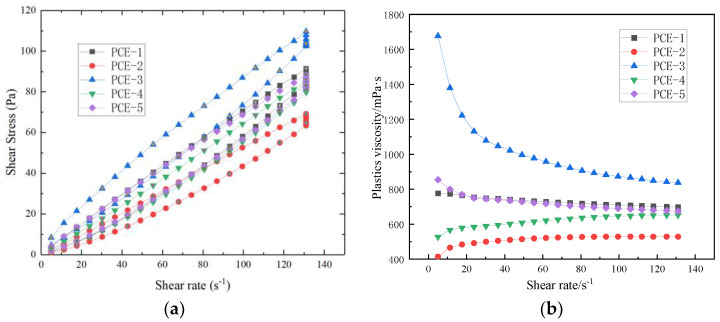
Effect of PCE onrheological properties of cement paste: (**a**) shear stress vs. shear rate; (**b**) plastic viscosity vs. shear rate.

**Figure 12 materials-16-04823-f012:**
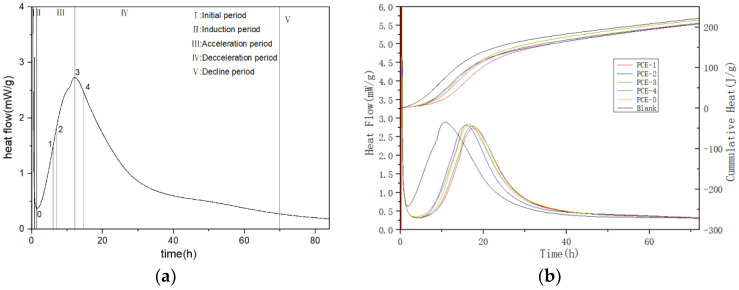
Heat curve of cement hydration: (**a**) the five stages of cement hydration; (**b**) hydration heat curve and accumulation heat curve of PCE.

**Figure 13 materials-16-04823-f013:**
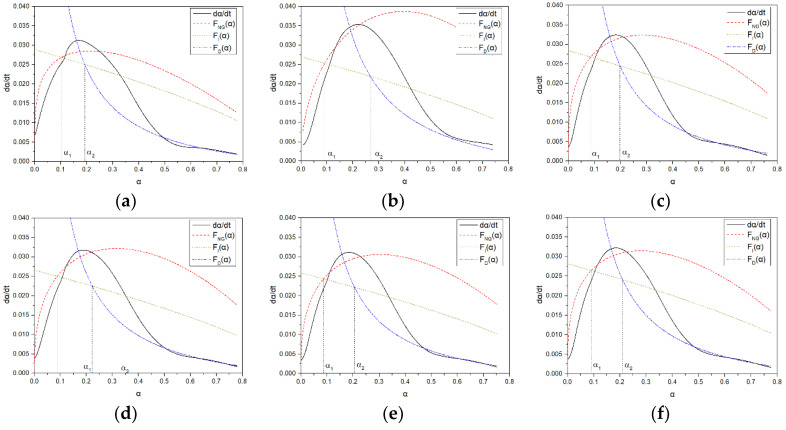
Hydration kinetics curve of PCE: (**a**) without PCE; (**b**) PCE-1; (**c**) PCE-2; (**d**) PCE-3; (**e**) PCE-4; (**f**) PCE-5.

**Table 1 materials-16-04823-t001:** Nonlinear forms of isotherm and kinetic models [9].

Adsorption Isotherm		Kinetic	
Isothermal Models	Names	Kinetic Models	Names
$q_{e}=\frac{q_{m}k_{L}C_{e}}{1+k_{L}C_{e}}$ $R_{L}=\frac{1}{1+k_{L}C_{0}}$	Langmuir	$q_{t}=q_{e}\left(1-e^{-k_{1}t}\right)$ $h_{0}=k_{1}q_{e}$	Pseudo-first order
$q_{e}=k_{F}C_{e}^{1/n_{F}}$	Freundlich	$q_{t}=\frac{k_{2}{q_{e}}^{2}t}{1+k_{2}q_{e}t}$; $h_{0}=k_{2}{q_{e}}^{2}$	Pseudo-second order
$q_{e}=\frac{RT}{b_{T}}In\left(k_{T}C_{e}\right)$	Temkin	$q_{t}=\frac{1}{b}In(1+abt)$	Elovich
$\varepsilon =RTIn\left(1+\frac{1}{C_{e}}\right)$ $E=\frac{1}{\sqrt{2k_{D}}}$	Dubinin-Radushkevich (D-R)	$q_{t}=k_{id}\sqrt{t}+C_{i}$	Intraparticle diffusion

**Table 2 materials-16-04823-t002:** Phase composition and physical properties of the cement.

CaO/%	SiO_2_/%	Al_2_O_3_/%	Fe_2_O_3_/%	MgO/%	SO_3_/%	Na_2_O/%	f-CaO/%	Loss
63.57	20.58	4.97	3.76	2.29	2.00	0.53	0.75	1.40

**Table 3 materials-16-04823-t003:** Monomer molar ratio for PCE synthesis.

Sample		Molar Ratio				
	n (TPEG)	n (AA)	n (MAA)	n (SMAS)	n (AMPS)	n (IA)
PCE-1	1	3.70	\	\	\	\
PCE-2	1	2.96	0.74	\	\	\
PCE-3	1	2.96	\	0.74	\	\
PCE-4	1	2.96	\	\	0.74	\
PCE-5	1	2.96	\	\	\	0.74

**Table 4 materials-16-04823-t004:** The molecular structure parameters of the PCE.

Sample	Mn	Mw	Conversion/%	PDI	Rh	Mark-Houwink α	D_AA_	L_bb_
PCE-1	25,247	44,843	89.60	1.78	4.96	0.26	0.81	18.16
PCE-2	25,132	47,278	76.00	1.88	5.41	0.23	0.83	19.03
PCE-3	11,306	17,262	73.60	1.53	3.46	0.16	0.80	6.75
PCE-4	39,908	74,641	79.90	1.87	6.47	0.21	0.79	28.65
PCE-5	21,941	38,915	77.50	1.77	4.88	0.19	0.99	15.39

**Table 5 materials-16-04823-t005:** Fit results of the PCE isotherm adsorption equation.

Sample	Langmuir			Freundlich			∆G/KJ·mol^−1^
	q_m_/mg·g^−1^	k_L_/min^−1^	R^2^	k_F_/(mg·g^−1^) (L·mg^−1^)^1/n^	n_F_	R^2^	
PCE-1	1.6066	4.0312	0.9985	1.2543	5.2680	0.9997	−26.6898
PCE-2	1.9621	2.6388	0.9937	1.3787	3.8467	0.9993	−27.5204
PCE-3	1.5137	2.9339	0.9957	1.0952	4.1491	0.9991	−25.4352
PCE-4	1.4914	0.8288	0.9956	1.0970	2.0369	0.989	−26.7894
PCE-5	1.5089	1.6562	0.9444	0.8439	2.4274	0.9748	−25.6087

**Table 6 materials-16-04823-t006:** Fit results of adsorption Kinetics of PCE.

Sample	PFO			PSO			I-D		
	q_e_/mg·g^−1^	k_2_/min^−1^	R^2^	q_e_/mg·g^−1^	k_2_/min^−1^	R^2^	k_id_/mg·g·min^−1^	C_i_	R^2^
PCE-1	2.8231	0.0658	0.9313	3.1199	0.0317	0.9549	0.2142	0.6526	0.8369
PCE-2	3.5192	0.0505	0.9791	3.9290	0.0194	0.9939	0.2637	0.7620	0.8545
PCE-3	3.7720	0.1134	0.9809	3.9377	0.0702	0.9878	0.2556	1.2226	0.6875
PCE-4	3.8058	0.1147	0.9810	3.9874	0.0654	0.9894	0.2624	1.1987	0.7107
PCE-5	2.3817	0.0784	0.9278	2.6097	0.0454	0.9502	0.1796	0.5859	0.8174

**Table 7 materials-16-04823-t007:** Fit results of PCE rheological.

Sample	Thixotropic Ring Area Pa/(s·cm³)	Bingham			H-B			
		μ/(mPa·s)	τ/MPa	R^2^	k	τ/MPa	n	R^2^
PCE-1	321.25	0.6926	1.6269	0.9996	0.9455	−0.4938	0.93915	0.9999
PCE-2	233.23	0.5369	0.9361	0.9999	0.5229	−0.8053	1.0052	0.9997
PCE-3	316.84	0.7873	8.5887	0.9972	1.7787	1.587	0.8419	0.9998
PCE-4	222.34	0.6646	2.0197	0.9995	0.4743	−0.0662	1.0665	0.9997
PCE-5	266.13	0.6649	2.4009	0.9993	1.0099	−0.3914	0.9184	0.9999

**Table 8 materials-16-04823-t008:** The hydration parameters of PCE on the cement particles.

Sample	t_0_/h	q_0_ (mW/g)	Q_0_ (J/g)	t_1_/h	t_2_/h	K_2_ (mW/(g∙h))	q_2_ (mW/g)	t_3_/h	t/h	q (mW/g)	Q/J.g^−1^
Blank	1.84	0.64	5.01	6.67	5.61	0.32	1.54	10.80	9.72	2.9	75.27
PCE-1	4.2	0.33	5.27	10.96	13.11	0.34	1.55	17.69	16.65	2.73	84.29
PCE-2	3.99	0.32	5.96	8.53	11.04	0.35	1.52	15.83	12.35	2.82	93.51
PCE-3	3.72	0.35	5.75	9.15	11.96	0.33	1.67	16.43	13.77	2.85	96.42
PCE-4	4.25	0.32	5.82	9.67	12.67	0.34	1.55	17.42	13.88	2.79	98.18
PCE-5	4.26	0.33	6.24	9.15	11.78	0.37	1.51	16.59	13.12	2.80	96.41

**Table 9 materials-16-04823-t009:** Kinetic parameters determined by means of the mathematical hydration model “Kinetic Analysis”.

Sample	R^2^	Q_max_/J·g^−1^	N_G_		I	D	α_1_	α_2_	α_3_	Δα_1_	Δα_2_
			K_NG_/h^−1^	n	K_I_/μm·h^−1^	K_D_/μm^2^·h^−1^					
Blank	0.9986	333.33	0.039	1.32	0.0096	0.0013	0.02	0.1	0.19	0.09	0.18
PCE-1	0.9961	277.78	0.045	2.01	0.009	0.0018	0.02	0.09	0.27	0.07	0.25
PCE-2	0.9999	312.5	0.044	1.5	0.0095	0.0013	0.02	0.09	0.2	0.07	0.18
PCE-3	0.9995	322.58	0.042	1.61	0.0089	0.0014	0.02	0.09	0.22	0.07	0.2
PCE-4	0.9995	322.58	0.040	1.57	0.0086	0.0013	0.02	0.09	0.21	0.07	0.19
PCE-5	0.9998	312.5	0.042	1.49	0.0094	0.0014	0.02	0.09	0.32	0.07	0.19

## Data Availability

Supporting data could be made available upon reasonable request.

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
