# Peer review of "Study on the Effect of Main Chain Molecular Structure on Adsorption, Dispersion, and Hydration of Polycarboxylate Superplasticizers"

_materials, 2023, doi:10.3390/ma16134823_

Round 1

Reviewer 1 Report

The purpose of this paper is to study the effect of the PCE molecular structure on the behaviour of some important properties for cement mixtures behaviour, such as adsorption, dispersion and hydration. The topic is interesting and timely and is worth publishing in this journal. Concrete admixtures are a field in permanent evolution, following the progress in chemistry and testing. The state of the art is correct an objective. The experimental program is properly designed, with an appropriate number of case studies and assessed properties. The conclusions are supported in the results and the discussion also provides in general reasonable explanations to support the obtained results. The overall structure of the paper is appropriate. The English writing could be improved, although not compromising the understanding of the content. As consequence, in my opinion this paper should be accepted after considering the following remarks. More detailed comments are given below:

·         Rows 30-31: improve this sentence, relativizing the benefits of admixtures. In particular, I disagree with the expression “the most effective”, making other very important elements secondary, like mix-design, amount of binding material, etc.

·         Row 107: authors suggest that the sharp valley shape of the induction period of the OPC hydration heat curve is detrimental; please explain why it is detrimental; it can also be beneficial in some situations where early age strength is important.

·         Rows 284-285: not completely accurate for PCE-4

·         Rows 299-301: improve explanation in the paper for the two simultaneous effects caused by carboyl groups (faster adsorption and smaller saturation adsorption capacity).

·         Row 501: add a previous presentation of the three hydration processes. NG process is explained after, D process is insufficiently explained also after and I could not find description of process I.

·         Row 512: improve the accuracy of the sentence (crystallization of C-S-H?)

·         Rows 30-31: replace “reducing” by “reduce” and “improving” by “improve”.

·         Row 97: stages I and II are both termed by “induction”.

·         Row 107: replace “sharper” by “sharp”;

·         Figure 1: write next to the images the corresponding PCE

·         Table 4: improve caption

·         Row 369: Figure number incorrect

·         Row 429: stages I and II are both termed by “induction”

·         Row 430: name of stage V is not the same in the text and in the figure

·         Rows 433-434: improve this description, since it is very confusing and inconsistent with the figure

·         Row 450: replace “long” by “longer”

·         Row 463: Figure number incorrect

·         Row 495: Figure number incorrect

Author Response

We are pleased to answer the questions of the editor and reviewers and the manuscript (materials-2411087) has also been extensively revised according to the comments.

Reviewer 1:Question 1: Improve this sentence, relativizing the benefits of admixtures. In particular, I disagree with the expression “the most effective”, making other very important elements secondary, like mix-design, amount of binding material, etc.

Answer:

Thanks very much for the editor’s question and we apology for the mistake. we have revised the wording of the sentence in question to enhance its accuracy.

Question 2: Authors suggest that the sharp valley shape of the induction period of the OPC hydration heat curve is detrimental; please explain why it is detrimental; it can also be beneficial in some situations where early age strength is important.

Answer:

Thanks a lot for the reviewer’s question. We acknowledge that in certain situations, such as during the preparation of shotcrete, there is a requirement to shorten the concrete setting time. However, in most cases, it is necessary for the concrete to have a certain period of induction time, allowing it to maintain workability during transportation to the construction site.

Question 3: Rows 284-285: not completely accurate for PCE-4

Answer:

Thanks a lot for the reviewer’s question. we have revised the wording of the sentence in question to enhance its accuracy.

Question 4: Rows 299-301: improve explanation in the paper for the two simultaneous effects caused by carboyl groups (faster adsorption and smaller saturation adsorption capacity).

Answer:

Thanks a lot for the reviewer’s question. we have revised the wording of the sentence in question to enhance its accuracy.

Question 5: Row 501: add a previous presentation of the three hydration processes. NG process is explained after, D process is insufficiently explained also after and I could not find description of process I.

Answer:

Thanks a lot for the reviewer’s question. We sincerely apologize for inadvertently deleting the corresponding section in the introduction during the revision process. We have now rectified this oversight by incorporating the relevant statements regarding hydration kinetics back into the introduction.

Question 6: Row 512:improve the accuracy of the sentence (crystallization of C-S-H?)

Answer:

Thanks a lot for the reviewer’s question. we have revised the wording of the sentence in question to enhance its accuracy. We have listed the sentence below: The larger KNG is, the easier the reaction occurs, resulting in a significant increase in the rate of formation of C-S-H and Ca(OH)2 crystals.

Question 7Comments on the Quality of English Language:

Answer:

Thanks a lot for the reviewer’s question. We sincerely apologize for the numerous mistakes that occurred due to our English proficiency. We have made the necessary corrections to rectify these errors.

Finally, thanks a lot for the editor and reviewers’ patience. We wish the modification can satisfy the editor and reviewers.

Reviewer 2 Report

This paper examines the impact of the molecular structure of the main chain on the adsorption, dispersion, and hydration of polycarboxylate superplasticizers. Infrared spectroscopy and gel chromatography were used to characterize the molecular structure, whereas a total organic carbon analyzer, rheometer, and isothermal microcalorimeter were utilized to investigate the adsorption, dispersion, and hydration properties. This article is particularly interesting because it describes the interactions between molecular structures and the isothermal adsorption model, adsorption kinetics, rheological model, and hydration kinetics of cement slurry systems. I have no substantive comments to make about the paper's content, as it has been presented with a high level of scientific rigor and contains comprehensive experimental test results (testing methods) and a systematic and high-quality discussion. To enhance this paper, it is suggested that the most recent references be utilized.

Author Response

Thank you for your valuable suggestions. We have checked the dates of the references and added several recent articles. The titles of the added references are as follows:

  1. Han, J. Study on the Hydration Properties of MultiComponent Composite Cementitious System with Low Water-binder Ratio. Master’s Thesis, Xinjiang Agricultural University, Urumqi, China, 2021.
  2. Han, F.H. Study on the Hydration Characteristics and Kinetics of Composite Binder. Ph.D. Thesis, China University of Mining and Technology, Beijing, China, 2015.
  3. Li, Z.P. Study on Optimization Design and Hydration Performance of Composite Cementitious Materials. Ph.D. Thesis, HarbinInstitute of Technology, Harbin, China, 2021.
  1. Krstulovi´c, R.; Dabi´c, P. A conceptual model of the cement hydration process. Cem. Concr. Res. 2000, 30, 693–698.
  2. Zhang, H.; Yang, Z.H.; Su, Y.F. Hydration kinetics of cement-quicklime system at different temperatures. Thermochim. Acta 2019, 673, 1–11.

Reviewer 3 Report

The authors present an interesting work on the effect of plasticiser polymer (PCE) structure on dispersion and hydration of cement. In the reviewer's opinion, this is a relevant work, which provides interesting findings, that deserves to be shared with the scientific community. However, and in contrast to the extensive scientific efforts, the presentation of the manuscript does not comply with the standards of a publication. Therefore, the reviewer suggests the authors to prepare a corrected version by carrying out an extensive edition based on the recommendations provided below in Comments to Author section. The suggestions and comments should be taken into account before accepting the article for publication. I am willing to do a re-review of the revised version.

1. The authors claim some interesting findings that: 

- PCE sulfonic acid groups reduces initial adsorption but accelerate the adsorption rate on cement and increase the adsorption amount at adsorption equilibrium.

- PCE acrylamide groups are beneficial to the initial dispersion and reduce the plastic viscosity of cement slurry, but prolong the induction period of cement hydration, while the introduction of sulfonic acid groups is not conducive to its retarding effect.

2. However, conclusions are not supported by results, as no statistical and measurement uncertinty was reported.

2.1 Please show the scatters all presented experimental results. This applies to Figures (via error bars) and also in the related Tables (by adding the +/- variabilities).

Results should include the measurement uncertainty, and expressed as mean value +/- (1x or 2x) standard deviation.

Errorbars are particularly needed in following Figures:

Figure 4. Charge density of PCE

Figure 6. The adsorption amounts of PCE at different concentrations

Figure 7. Adsorption kinetics curve of PCE on cement particles

Figure 9. Influence of PCE on the dispersion of cement pastes

Figure 11. Effect of PCE on rheological properties of cement paste: (a)Fig. 11 The shear stress vs 400

shear rate:; (b)Fig 12 The plastic viscosity vs shear rate 

2.2 Please discuss the differences between the results considering the statistical significance (e.g. using ANOVA, t-test).

For example this is in particular needed to discuss the changes in main results.

2.3 Conclusions should state if the observed effects are statistically significant (using quantitative statistical indicators) or not!

3. Overall, the experimental methods used are well known and the results are not explained in detail. It seems many experiments have been carried out and their results are summarized without in-depth discussion and comparison with other studies on the same subject area. 

Prospects, challenges, future work, limitations, etc. must be discussed in more depth.

4. Please also add future research steps which will follow this work.

5. Use of many abbreviations makes difficult to follow the text.

Please provide a table listing all abbreviations and parameters.

7. The model predictions should be clearly evaluated/validated by measurement results. Calibration of the model should be discussed in detail. Some sensitivity study could be added as well.

English language quality should in general be improved. It looks like the manuscript is writen in a hurry, having many gramatical and editorial mistakes. 

E.g. check caption of the fir 11:

'Figure 11. Effect of PCE on rheological properties of cement paste: (a)Fig. 11 The shear stress vs 400 shear rate:; (b)Fig 12 The plastic viscosity vs shear rate '

also, the title of section should use only acronims, e.g.:

3.1.3 GPC 

Author Response

We are pleased to answer the questions of the editor and reviewers and the manuscript (materials-2411087) has also been extensively revised according to the comments.

Reviewer 3:

Question 1: 2.1 Please show the scatters all presented experimental results. This applies to Figures (via error bars) and also in the related Tables (by adding the +/- variabilities).

Results should include the measurement uncertainty, and expressed as mean value +/- (1x or 2x) standard deviation.
Answer:

Thanks a lot for the reviewer’s question. Figures 4, 6, 7, and 9 have been updated to include error bars as requested. The modified figures are shown below. In Figure 11, the rheological data exhibits significant variation in shear stress and plastic viscosity with shear rate, and the data points are numerous. Therefore, we believe that adding error bars in this case would not be meaningful.

Figure 4. Specific charge density of PCE      Figure 6. The adsorption amounts of PCE at different concentrations

Figure 7. Adsorption kinetics curve of PCE on cement particles   Figure 9. Influence of PCE on the fludity of cement pastes

Question 2: How they are explain small, not significant, influence of PCE on viscosity reduction on cement paste, Figure 5?

Answer:

Thanks a lot for the reviewer’s question. We appreciate the comments from the reviewer. First, I would like to apologize for the confusion in our diagrams that may have made the changes in viscosity appear inconspicuous. We will clarify this issue by reconstructing the figures (Figure 3) based on the impacts of different micro-structured PCEs on viscosity. Given that the proportion of the microstructure changes we implemented is quite small, the viscosity changes in some of the cement slurries may seem subtle or even quite similar. We thank you for your insightful observations, and we will make sure to address these concerns in our revision. We hope it can satisfy the reviewer.

Question 3: 2.2 Please discuss the differences between the results considering the statistical significance (e.g. using ANOVA, t-test).

Answer:

Thank you for the valuable comments provided by the reviewer. We have employed a repetition testing approach to verify the experimental results and have utilized multiple mathematical models for fitting, leading to the conclusions presented in this paper. Therefore, we believe that the obtained conclusions hold statistical significance.

Question 4: 2.3 Conclusions should state if the observed effects are statistically significant (using quantitative statistical indicators) or not!

Answer:

Thank you for the valuable comments provided by the reviewer. We have incorporated the suggested changes and included additional data analysis in the conclusion to support our findings. However, it is important to note that this study primarily focuses on the qualitative investigation of the impact of different functional groups on PCE performance. The effect of varying dosage on performance has not been specifically studied, making it challenging to provide conclusive evidence through explicit data analysis.

Question 5: Please also add future research steps which will follow this work.

Answer:

Thank you for the reviewer's inquiry. In our future work, we plan to conduct a quantitative study on the impact of varying dosages of different functional groups on PCE performance. We aim to investigate the optimal molar amount of functional groups that can enhance PCE performance, which will provide valuable insights for subsequent product development.

Question 5: Use of many abbreviations makes difficult to follow the text.Please provide a table listing all abbreviations and parameters.

Answer:

Thank you for the valuable feedback. We have compiled a table 1 containing all the abbreviations used throughout the manuscript. Please review it and let us know if any revisions are needed.

Table 1. abbreviations and parameters.

Abbreviation

Full name

pce

Polycarboxylate superplasticizer、Polycarboxylate ether

FCPS

fresh cement pastes

CH

calcium hydroxide

C3S

tricalcium silicate

C-S-H

Calcium Silicate Hydrate。

AFT

Thiosulfoaluminate

AFM

Monosulfoaluminate

AA

Acrylic acid

MAA

methacrylic acid

SMS

sodium methacrylate sulfonate

AMPS

2-acrylamide-2-methylpropane sulfonic acid

IA

itaconic acid

APS

Ammonium persulfate

SH

Sodium hexametaphosphate

SCD

Specific charge density

TOC

total organic carbon

FI-TR

Fourier Transform Infrared Spectroscopy

GPC

Gel permeation chromatography

NG

nucleation and crystal growth

I

 interfacial reaction

D

diffusion

Question 6: The model predictions should be clearly evaluated/validated by measurement results. Calibration of the model should be discussed in detail. Some sensitivity study could be added as well.

Answer:

Thank you for the valuable feedback. The models we used in this study, such as Langmuir, Freundlich, and Dubinin-Radushkevich models, are commonly employed in this field of research. These models have been widely used due to their high fitting coefficients, typically exceeding 0.99. Therefore, we did not perform any model modifications. We kindly ask the reviewing editor for understanding in this matter.

Question 7Comments on the Quality of English Language:

Answer:

Thanks a lot for the reviewer’s question. We sincerely apologize for the numerous mistakes that occurred due to our English proficiency. We have made the necessary corrections to rectify these errors.

Finally, thanks a lot for the editor and reviewers’ patience. We wish the modification can satisfy the editor and reviewers.

Reviewer 4 Report

An interesting paper with one area for expansion.  It would be good to have a summary of how these compare with traditional reference materials - there are no such comparisons in the discussion.  this leaves the conclusions interesting but unclear.

The English is good if a little formal at times - it is clear however.

Author Response

Thank you for your valuable suggestions. We indeed overlooked the comparison in this aspect in the introduction. Following your advice, we have added a comparison of the current industry research status in the last paragraph of the introduction.

Reviewer 5 Report

The paper's topic is exciting and up-to-date. Superplasticisers for concrete, based on polycarboxylic ethers, are the top group of modern fluidising admixtures and yet the specific mechanisms of their action still need to be fully understood. The paper brings new information about the influence of molecular structure on the admixture's performance. The Authors' approach to the problem is to analyse the results of numerous tests covered by the broad experimental program. The text is generally well prepared, the presentation is clear, and the conclusions are logical and justified.

I suggest two corrections to the text; first, the sentence below Table 1 ("After the adsorption…") is not understandable – grammar and syntax need to be improved. Second, why are two induction periods listed in the second paragraph on page 3?

With the above amendments, I recommend publishing the paper.

Generally English language is fine for me, but the sentence below Table 1 ("After the adsorption…") is not understandable – grammar and syntax need to be improved.

Author Response

Thank you for your valuable suggestions. We apologize for the issues in English grammar and writing details. We have made the necessary modifications based on your feedback. Sentence 1 has been revised to "After different molecular structures of PCE are adsorbed onto the surface of cement particles or hydration products...". Additionally, in the third page, we have changed the first occurrence of "induction" to "initial". Thank you once again for your careful reading and valuable feedback.